

# Bacterial production of polyhydroxyalkanoates (PHAs) using various waste carbon sources

Aansa Naseem[1], Ijaz Rasul[1], Zulfiqar Ali Raza[2], Faizan Muneer[1,3], Asad ur Rehman[4] and Habibullah Nadeem[1]

[1] Department of Bioinformatics and Biotechnology, Government College University, Faisalabad, Punjab, Pakistan
[2] Department of Applied Sciences, National Textile University, Faisalabad, Punjab, Pakistan
[3] Biological Oceanography Lab, National Institute of Oceanography, Karachi, Sindh, Pakistan
[4] Dr Ikram-ul-Haq Institute of Industrial Biotechnology, Government College University Lahore, Lahore, Punjab, Pakistan

Corresponding author
Habibullah Nadeem,
habibullah@gcuf.edu.pk

## ABSTRACT

Synthetic plastics are in great demand in society due to their diversified properties, but they cause environmental pollution due to their non-biodegradable nature. Therefore, synthetic plastics are in need to be replaced with biodegradable plastics. Polyhydroxyalkanoates (PHAs), bacterial biopolymers are natural alternative to synthetic plastics. These are present inside the bacterial cytoplasm in granular form. Presently, the production cost of PHA is high due to expensive carbon substrates used in its biosynthesis. Therefore, this study focuses on the cost-effective production of PHA using waste carbon sources. Rice bran and sugarcane molasses were used as the carbon source for PHA production from *Bacillus subtilis*, *Bacillus cereus*, *Alcaligenes* sp. and *Pseudomonas aeruginosa*. PHA production from these bacterial strains was confirmed through Sudan Black-B screening. With rice bran, as carbon source, the highest PHA yield obtained was for *P. aeruginosa*, which yielded 93.7% and lowest was 35.5% for *B. cereus*. Surprisingly, *B. cereus* produced the highest cell dry mass (0.045 g/L) but its extracted PHA contents were lowest being only 0.02 g/L. *Alcaligenes* sp. with 0.031 g/L CDM yielded 87.1% PHA. *B. subtilis* had a CDM 0.029 g/L, 0.02 g/L PHA content and a yield of 69.10%. In the case of sugarcane molasses, *P. aeruginosa* produced 95% PHA yield, 0.02 g/L CDM, and 0.019 g/L PHA content. *Alcaligenes* sp. yielded 90.9% PHA, 0.011 g/L CDM, and 0.01 g/L PHA content. *B. subtilis* produced 91.6% PHA yield, 0.012 g/L CDM, 0.011 g/L PHA content; *B. cereus* produced 80% PHA yield, 0.015 g/L CDM, 0.012 g/L PHA content at 37 °C, pH 7. Higher concentrations of carbon sources increased the CDM and decreased the PHA yield. The maximum yield of PHA was obtained from sugarcane molasses. 24–48 h of incubation was optimal for *B. subtilis* and *B. cereus,* while for *Alcaligenes* and *P. aeruginosa* incubation time of 48–96 h was desirable for higher PHA yield. The extracted biopolymers were analyzed by Fourier transform infrared spectroscopy (FTIR), which identified the extracted biopolymers as poly-3-hydroxybutyrate P(3HB). The thermal properties of the extracted biopolymers, such as melting temperatures, were analyzed by differential scanning calorimetry (DSC), which confirmed the thermal stability.

# INTRODUCTION

Synthetic plastics are non-biodegradable and cause environmental pollution due to their toxic nature (*Angra, Sehgal & Gupta, 2023*). Therefore, there is an urgency to search for sustainable and ecofriendly alternatives to modern day plastics. For the purpose, synthetic plastics are being replaced by natural, eco-friendly, biodegradable polymers such as lignin, starch, chitin, and microbial polysaccharides and polyesters (*Muneer et al., 2021b*). These biopolymers have mechanical and physicochemical similarities to synthetic plastics (*Amaro et al., 2019*). The use of biopolymers due to their similarities with synthetic plastics is highly encouraged for a sustainable environment and biosphere, moreover these can be used in a wide range of industrial and pharmaceutical applications (*Vicente, Proença & Morais, 2023*). Bioplastics can be produced from various sources like plants and microorganisms (*Takagi et al., 2004*).

Several diverse groups of sustainable polyesters can be used as biomaterials for the sustainable production of plastic products; one such group of polyester is known as polyhydroxyalkanoates (PHAs), which is synthesized by microorganisms (*Qiang et al., 2018*). PHA is a natural alternative to synthetic plastics that is biodegradable and non-toxic (*Mannina et al., 2019*). PHA is the most commonly studied biodegradable polymer. More than 300 species of bacteria and archaea are used to synthesize PHA naturally (*Khatami et al., 2021*). It is a reservoir of carbon and energy accumulated in bacteria which is produced by microorganisms under nutrient stress of nitrogen, sulfur, zinc, phosphorous, potassium, magnesium, iron, and oxygen (*Masood, Yasin & Hameed, 2014*). It is usually present as granular structure inside the cytoplasm of the microbial cells (*Aljuraifani, Berekaa & Ghazwani, 2019a*). These granules are round-shaped and are 0.1–0.2 μm in diameter. These inclusion bodies perform the function of storage in the vegetative cells of the bacteria (*Marang, Van Loosdrecht & Kleerebezem, 2018*). The formation of such granules of PHAs varies among different microbes, (*Zhang et al., 2018*). Microbes containing PHA granules can survive under a scarcity of food and face environmental challenges (*Slaninova et al., 2018*; *Obruca et al., 2017*).

At present, the production cost of PHA is too high, which is the major limitation of its industrial scale production. Usually, the use of expensive substrates leads to the higher cost of PHA production (*Guerra-Blanco et al., 2018*). As a potential solution, cost-effective carbon sources should be used (*Raza, Abid & Banat, 2018*). Readily available and sustainable carbon sources should be used for the sustainable production of PHA (*Koller et al., 2017*; *Kourmentza et al., 2017*). Using waste materials as carbon sources is another solution to make production cost-effective (*Muiruri et al., 2023*). A large amount of waste feedstock is available in the environment, which can be utilized for PHA production (*Sagastume et al., 2016*). Pretreated and well-characterized waste material is used for effective PHA production (*Rodriguez et al., 2018*). The carbon of pretreated

waste material is higher than untreated waste material and supports microbial growth (*Pittmann & Steinmetz, 2016*). The pre-treatment methodology increases the dilution of organic waste, removes solid materials from it, controls temperature and pH, and increases waste materials' sterility (*Amulya et al., 2016*; *Basset et al., 2016*). The type of carbon source affects the structure and style of the PHA monomers.

Along with carbon sources, bacterial cell growth and PHA content are also affected by the carbon and nitrogen (C/N) ratio (*Cui, Shi & Gong, 2017*). The most common nitrogen supplements include $NH_4Cl$, $(NH_4)_2SO_4$, and $NH_4NO_3$. Different C:N ratios have increased the production of microbial biomass and PHA content (*Muneer et al., 2022a*). Hence, there is a direct relation between accumulated PHA and C:N ratio, while there is an inverse relation between PHA accumulation and microbial growth (*Ahn, Jho & Nam, 2015*; *Cui, Shi & Gong, 2017*). Various microorganisms can produce PHA but studies have shown bacterial PHA production as more economical because of its capacity to accumulate high contents of PHA (*Verlinden et al., 2007*). According to various studies, the best incubation period for PHA production is 24–96 h; the best pH is 7 with a temperature of 37 °C for different bacterial species. After the growth of microbial biomass under optimized conditions, extraction and purification of microbial PHA are essential steps because purified PHA is further used to determine the physical and chemical characteristics of produced PHA. To cope with the cost and for sustainable production of PHA, waste biomass can be used as a carbon source for PHA production. Hence, cost-effective PHA can replace petroleum-based plastics economically (*Muneer et al., 2020*). Therefore, the present study aimed to optimize polyhydroxyalkanoates (PHA) production by using rice bran and sugarcane molasses as the carbon sources under optimized conditions of bacterial growth and to achieve sustainable development goals (SDGs) by creating a plastic-free environment to save the global biodiversity and ecosystem from the hazardous effects of the plastic pollution (*Muneer et al., 2021a*).

## MATERIALS AND METHODS

### Microorganisms and chemicals

Four bacterial strains *Bacillus subtilis* (accession number KM282517), *Bacillus cereus* (accession number KM282518), *Pseudomonas aeruginosa* (*Maqbool et al., 2016*), *Alcaligenes* sp. used in this study were collected from Industrial Biotechnology Laboratory Government College University Faisalabad, Pakistan. These strains were reactivated and fresh cultures were prepared on nutrient agar plates. The inoculum was prepared by adding the bacterial cells to the nutrient broth. This inoculum was used for further studies. All the chemicals used in this study were of analytical grade, purchased from RDH (Germany), CarlRoth Gmbh (Germany), Sigma-Aldrich (USA), and Europroxima (The Netherlands).

### Bacterial screening for PHA production

Sudan Black-B staining was used for the screening of bacterial strains to confirm the production of PHA. Alcoholic Sudan Black-B (0.3%) was used for the bacterial staining, with some modifications as described by *Mohanrasu et al. (2020)*. Two methods were used for the bacterial screening with Sudan Black-B. In the first method, the agar plates with

the PHA accumulating strains and control strains were flooded with the prepared Sudan Black B solution for 30–45 min (*Alshehrei, 2019*). The second method was the smear slide method, in which the slides were observed under a microscope after staining with alcoholic Sudan Black-B solution (*Muneer et al., 2022*).

## Carbon sources

This study used rice bran and sugarcane molasses as cost-effective carbon sources. Rice bran was collected from the local rice milling point at Sheikhupura Road, Faisalabad, Pakistan. The sugarcane molasses was arranged from the Tandlianwala and Kanjwani Sugar Mills Limited, Faisalabad, Pakistan. Raw sugarcane molasses contains variable content of different sugars like glucose, fructose, and sucrose, which aid in PHA production. Some inhibitory metal ions are also found in cane molasses, which are unsuitable for PHA production; therefore, they were pretreated before its use.

## Pre-treatment of sugarcane molasses

Raw sugarcane molasses contains variable content of various carbohydrates like glucose, fructose, and sucrose, which aid in PHA production. Some inhibitory metal ions are also found in cane molasses, which are unsuitable for PHA production. Deionized distilled water was added to the concentrated sugarcane molasses to set the required sugar content. Activated charcoal was added to the diluted molasses, stirred for 1–2 h using a magnetic stirrer, and centrifuged at 6,000 rpm for 10 min. The pH of the filtered supernatant was set at 7–7.5 and autoclaved (*Jo et al., 2021*).

## PHA production media & culture conditions

The selected bacterial strains were cultured in shaking flasks containing minimal solution medium (MSM). Before inoculating into MSM, these bacterial strains were cultured in the nutrient broth (NB) medium containing: Peptone; 5 g/L, Yeast extract; 1.5 g/L, Beef extract; 1.5 g/L, NaCl; 5 g/L, glucose; 10 g/L, pH; 7, incubated at 37 °C, 150 rpm for 24 h. 10ml of each inoculum was added in autoclaved MSM. The composition of MSM was derived with some modifications by (*Gomaa, 2014*). MSM consisted of different salts in the content ratio of (g/L): $Na_2HPO_4.2H_2O$; 2.0, $NaHCO_3$; 0.5, $MgSO_4.7H_2O$; 0.5, $NH_4Cl$; 1.0, $CaCl_2.2H_2O$; 0.01, $KH_2PO_4$; 2.0, and some trace elements including $H_3BO_3$; 0.3, $ZnSO_4.7H_2O$; 0.08, $CoCl_2.6H_2O$; 0.2, $NiCl_2.6H_2O$; 0.02, $MnCl_2.4H_2O$; 0.03, $CuCl_2.2H_2O$; 0.01. Five different concentrations of rice bran (0.5%, 1.0%, 1.5%, 2.0%, and 2.5%) and pretreated sugarcane molasses (2.0%, 4.0%, 6.0%, 8.0%, and 10.0%) were added to the MSM medium in different flasks as a carbon source for PHA production. Three replicates were used during the experiment using each carbon source and for their various concentrations. The data obtained is given in Tables 1 and 2.

## Extraction and purification of PHA

PHA extraction was done using the previously reported methodology with some modifications (*Mohandas et al., 2018*; *Aljuraifani, Berekaa & Ghazwani, 2019b*). After the incubation period, the cell culture was collected and harvested. The cells of the culture containing rice bran and cane molasses were harvested differently. The biomass of the cell

Naseem et al. (2024), *PeerJ*, DOI 10.7717/peerj.17936

**Table 1  Optimization of rice bran concentration and Incubation time.**

| Bacterial strains | Incubation time (h) | Rice Bran % | | | | | | | | | | | | | | |
|---|---|---|---|---|---|---|---|---|---|---|---|---|---|---|---|---|
| | | 0.5 | | | 1.0 | | | 1.5 | | | 2.0 | | | 2.5 | | |
| | | CDM (g/L) | PHA (g/L) | PHA % | CDM (g/L) | PHA (g/L) | PHA % | CDM (g/L) | PHA (g/L) | PHA % | CDM (g/L) | PHA (g/L) | PHA % | CDM (g/L) | PHA (g/L) | PHA % |
| *B. subtilis* | 24 | 3.8 | 1.7 | 44.7 | 11.0 | 1.4 | 12.7 | 5.5 | 1.9 | 12.3 | 11.2 | 1.0 | 8.9 | 9.3 | 1.4 | 15 |
| | 48 | 2.9 | 2.0 | 69 | 7.0 | 0.5 | 7.1 | 3.2 | 1.1 | 8.3 | 11.9 | 1.1 | 9.2 | 14.7 | 1.3 | 8.8 |
| | 72 | 2.6 | 0.7 | 26.9 | 2.5 | 0.002 | 8 | 6.4 | 1.6 | 9.7 | 11.4 | 0.5 | 4.4 | 10.0 | 0.5 | 5 |
| | 96 | 1.6 | 0.5 | 31.3 | 11.9 | 0.8 | 6.7 | 3.7 | 1.2 | 8.8 | 11.1 | 0.9 | 8.1 | 12.8 | 0.8 | 6.3 |
| *B. cereus* | 24 | 4.5 | 1.6 | 35.5 | 6.3 | 1.3 | 20.6 | 8.7 | 1.0 | 11.5 | 10.1 | 1.0 | 9.9 | 5.6 | 0.6 | 10.7 |
| | 48 | 6.7 | 0.6 | 8.9 | 4.8 | 1.1 | 23 | 9.3 | 0.7 | 7.5 | 13.0 | 0.9 | 6.9 | 8.9 | 1.2 | 13.5 |
| | 72 | 7.6 | 0.4 | 5.2 | 9.8 | 0.8 | 8.2 | 4.6 | 0.8 | 5.5 | 19.8 | 2.1 | 10.6 | 8.5 | 1.5 | 17.7 |
| | 96 | 4.6 | 1.1 | 23.9 | 11.0 | 2.2 | 20 | 11.5 | 2.2 | 19.1 | 16.6 | 2.5 | 15 | 11.4 | 1.6 | 14 |
| *Alcaligenes* | 24 | 2.0 | 0.9 | 45 | 3.8 | 1.3 | 34.2 | 6.4 | 1.5 | 23.4 | 7.0 | 1.1 | 15.7 | 4.9 | 1.6 | 32.6 |
| | 48 | 2.4 | 1.7 | 70.8 | 3.7 | 1.7 | 45.9 | 4.8 | 3.3 | 68.7 | 6.4 | 2.0 | 31.2 | 6.5 | 2.1 | 32.3 |
| | 72 | 3.1 | 2.7 | 87.1 | 5.0 | 3.2 | 64 | 6.4 | 3.0 | 46.8 | 14.5 | 3.3 | 22.7 | 7.3 | 3.4 | 46.5 |
| | 96 | 2.6 | 0.3 | 11.5 | 4.1 | 0.8 | 19.5 | 6.0 | 0.2 | 3.3 | 8.9 | 1.0 | 11.2 | 8.6 | 1.0 | 11.6 |
| | 120 | 3.3 | 0.4 | 12.1 | 6.5 | 0.7 | 10.8 | 9.8 | 0.9 | 9.2 | 10.0 | 0.9 | 9 | 10.6 | 1.8 | 16.9 |
| *P. aeruginosa* | 24 | 2.9 | 1.0 | 34.4 | 2.6 | 1.1 | 42.3 | 5.1 | 1.2 | 23.5 | 5.3 | 1.1 | 20.7 | 6.9 | 1.1 | 15.9 |
| | 48 | 2.5 | 1.9 | 76 | 3.9 | 2.9 | 74.3 | 4.3 | 1.7 | 39.5 | 5.4 | 1.0 | 18.5 | 15.5 | 2.9 | 18.7 |
| | 72 | 3.2 | 3.0 | 93.7 | 5.2 | 2.6 | 50 | 4.6 | 3.8 | 82.6 | 21.7 | 3.4 | 15.6 | 10.0 | 3.0 | 30 |
| | 96 | 1.9 | 0.5 | 26.3 | 3.7 | 0.6 | 16.2 | 3.9 | 0.8 | 20.5 | 9.1 | 1.1 | 12 | 6.7 | 0.7 | 10.4 |
| | 120 | 2.7 | 0.5 | 18.5 | 6.1 | 0.6 | 9.8 | 7.3 | 0.1 | 1.4 | 7.9 | 0.5 | 6.3 | 8.7 | 0.8 | 9.2 |

Naseem et al. (2024), *PeerJ*, DOI 10.7717/peerj.17936

**Table 2** Optimization of Sugarcane molasses concentration and Incubation time.

| Bacterial strains | Incubation time (h) | Sugarcane molasses % | | | | | | | | | | | | | | |
|---|---|---|---|---|---|---|---|---|---|---|---|---|---|---|---|
| | | 2.0 | | | 4.0 | | | 6.0 | | | 8.0 | | | 10.0 | | |
| | | CDM (g/L) | PHA (g/L) | PHA % | CDM (g/L) | PHA (g/L) | PHA % | CDM (g/L) | PHA (g/L) | PHA % | CDM (g/L) | PHA (g/L) | PHA % | CDM (g/L) | PHA (g/L) | PHA % |
| | 24 | 12.0 | 1.1 | 91.6 | 0.3 | 0.1 | 33.3 | 1.5 | 0.9 | 60 | 1.0 | 0.2 | 20 | 0.3 | 0.2 | 66.6 |
| | 48 | 0.7 | 0.2 | 28.5 | 0.8 | 0.4 | 50 | 0.5 | 0.1 | 20 | 0.4 | 0.1 | 25 | 0.5 | 0.3 | 60 |
| *B. subtilis* | 72 | 0.6 | 0.1 | 16.6 | 0.7 | 0.1 | 14.2 | 0.4 | 0.2 | 50 | 1.1 | 0.5 | 45.4 | 0.3 | 0.1 | 33.3 |
| | 96 | 0.2 | 0.1 | 50 | 0.6 | 0.1 | 16.6 | 0.7 | 0.3 | 42.8 | 1.1 | 0.6 | 54.5 | 0.7 | 0.2 | 28.5 |
| | 24 | 1.5 | 1.2 | 80 | 0.8 | 0.4 | 50 | 0.6 | 0.1 | 16.6 | 0.9 | 0.3 | 33.3 | 0.3 | 0.2 | 66 |
| | 48 | 1.1 | 0.8 | 72.7 | 0.8 | 0.6 | 75 | 0.9 | 0.5 | 55.5 | 0.2 | 0.1 | 50 | 0.5 | 0.2 | 40 |
| *B. cereus* | 72 | 0.8 | 0.5 | 62.5 | 1.1 | 0.4 | 36.3 | 1.3 | 0.7 | 53.8 | 0.3 | 0.2 | 66.6 | 0.8 | 0.4 | 50 |
| | 96 | 0.7 | 0.4 | 57.1 | 0.5 | 0.1 | 20 | 0.5 | 0.2 | 40 | 0.4 | 0.1 | 25 | 0.5 | 0.2 | 40 |
| | 24 | 1.7 | 1.3 | 76.4 | 1.9 | 1.2 | 63.1 | 1.7 | 1.1 | 64.7 | 10.5 | 1.3 | 12.3 | 2.1 | 0.5 | 23.8 |
| | 48 | 2.0 | 1.0 | 50 | 3.0 | 2.1 | 70 | 2.2 | 1.6 | 72.7 | 2.0 | 1.1 | 55 | 1.5 | 0.9 | 60 |
| *Alcaligenes* | 72 | 1.2 | 0.2 | 16.6 | 1.5 | 1.0 | 66.6 | 1.3 | 0.7 | 53.8 | 1.8 | 1.4 | 77.7 | 2.2 | 0.4 | 18.1 |
| | 96 | 1.1 | 0.7 | 63.6 | 0.7 | 0.4 | 57.1 | 2.1 | 1.5 | 71.4 | 1.1 | 1.0 | 90.9 | 1.1 | 0.4 | 36.3 |
| | 24 | 1.0 | 0.5 | 50 | 0.8 | 0.5 | 62.5 | 0.6 | 0.1 | 16.6 | 1.7 | 0.6 | 35.2 | 2.0 | 0.4 | 20 |
| | 48 | 0.7 | 0.4 | 57.1 | 1.4 | 1.1 | 78.5 | 1.4 | 1.0 | 71.4 | 2.5 | 1.0 | 40 | 1.6 | 0.5 | 31.2 |
| *P. aeruginosa* | 72 | 2.3 | 2.1 | 91.3 | 3.6 | 2.5 | 69.4 | 4.0 | 1.6 | 40 | 4.8 | 2.8 | 58.3 | 4.3 | 3.3 | 76.7 |
| | 96 | 2.0 | 1.9 | 95 | 3.5 | 2.3 | 65.7 | 3.7 | 1.7 | 45.9 | 4.8 | 3.0 | 62.5 | 5.9 | 4.3 | 72.8 |

culture containing the rice bran was collected. An equal volume of sodium hypochlorite and chloroform was added to the biomass and incubated at 37 °C for 1 h. After incubation, the mixture was at 8,000 rpm for 15 min, discarding the upper phase containing cellular digestions. The lower chloroform phase was collected using a sterile syringe to collect biopolymers by adding 10 volumes of ice-cold methanol. Then, the precipitated PHA was dried at 65 °C for 24 h. The biomass of the sugarcane molasses cell culture was collected, washed with distilled water, and lyophilized by adding 40% sodium hypochlorite, vortexed vigorously, and incubated at 37 °C for 1 h. After incubation, centrifuged the mixture at 8,000 rpm for 15 min, discarded the supernatant, washed twice with distilled water, and then splashed the pellet with acetone and methanol. Centrifuged at 10,000 rpm for 5 min, suspended the pellet in boiling chloroform, and dried at 65 °C for 24 h. The formula in Eq. (1) is used to calculate the PHA content of the extracted biopolymer.

$$PHA(\%, w/w) = \frac{\text{Weight of PHA(g/L)}}{\text{CDM(g/L)}} \times 100. \tag{1}$$

## Optimization of the process parameters

To optimize the different parameters for PHA hyperproduction, other culture conditions such as concentration of carbon source, incubation period, pH conditions, and temperature were selected, which helped to determine the optimum conditions for PHA production by *B. subtilis*, *B. cereus*, *Alcaligenes* sp., and *P. aeruginosa* respectively. The carbon sources of different concentrations were used to optimize the production of PHA. In the case of rice bran as a carbon source, five concentrations of rice bran 0.5%, 1.0%, 1.5%, 2.0%, and 2.5% were added in five separate Erlenmeyer flasks containing MSM media. In the case of sugarcane molasses as a carbon source, five concentrations of pretreated sugarcane molasses 2.0%, 4.0%, 6.0%, 8.0%, and 10.0% were added in five separate Erlenmeyer flasks containing MSM media. *Alcaligenes* sp. and *P. aeruginosa* were incubated for 120 h, and *B. subtilis* and *B. cereus* were incubated for 96 h in the shaking incubator. Cell dry mass (CDM) and PHA yield were determined at 24, 48, 72, 96, and 120 h along with the percentage yield of PHA derived from bacterial strains with both carbon sources. The pH of the bacterial growth medium (MSM) was optimized to determine the effect of pH on PHA production. The pH of the medium was varied from pH 4.0−9.0 for each bacterial strain by adding NaOH & HCl.

## Analytical characterization

Fourier transform infrared spectrophotometry (FTIR) was used to analyze the chemical structure of the extracted polyhydroxyalkanoates produced from *B. subtilis, B. cereus, Alcaligenes* sp., and *P. aeruginosa* in the form of granular powder (*Morya et al., 2021*). FTIR analysis was done by dissolving the PHA powder in the chloroform and KBr (potassium bromide) pellet. The spectra were recorded in the 600 to 4,000 cm$^{-1}$ range using FTIR spectrophotometer. The extracted PHA samples were further thermally characterized by a differential scanning calorimeter (TA instruments Trios V4.1) to determine the extracted PHA's melting temperature (Tm). The heating temperature was adjusted from

25−350 °C, to check the melting temperature ($T_m$) of the extracted samples under the nitrogen atmosphere at the heating rate of 10 °C per minute.

## Statistical analysis

These experiments were performed in triplicate and the obtained data was statistically analyzed with one-way ANOVA. The *p*-values were calculated. Significant, and non-significant values were noted.

## RESULTS

### Bacterial screening for PHA production

*B. subtilis*, *B. cereus*, *Alcaligenes* sp., and *P. aeruginosa* strains were grown on freshly prepared nutrient agar plates from the stock cultures. All the strains showed good growth (Fig. 1A) on the plates and were found satisfactory for further study. These bacterial strains were stained with Sudan Black-B. The results (Fig. 1B) indicated the production of PHA in all four bacterial strains (*Shah, 2014*). Under the compound microscope, the black granules (Fig. 1C) represented the Sudan Black-B positive isolates for PHA production.

### Extraction and purification of PHA biopolymer

PHA polymer was extracted by using sodium hypochlorite and chloroform digestion from cellular biomass (Fig. 2). PHA production from *B. subtilis*, *B. cereus*, *Alcaligenes* sp., and *P. aeruginosa* was analyzed after different time intervals, with rice bran and sugarcane molasses as carbon source respectively. Under optimized conditions of rice bran, *P. aeruginosa* effectively utilized 0.5% rice bran and produced the maximum PHA yield, which was 93.7%, 3.2 g/L CDM, 3.0 g/L PHA content, after 72 h at 37 °C. According to previously reported studies, *Pseudomonas* sp. P (16) produced a 90.9% PHA yield (*Aljuraifani, Berekaa & Ghazwani, 2019b*), which has little difference from the research results presented in PHA production yield obtained from *Alcaligenes* sp. was 87.1%, 3.1 g/L CDM, and 2.7 g/L PHA content with 0.5% rice bran, after 72 h at 37 °C. PHA production yield obtained from *B. subtilis* was 69%, 2.9 g/L CDM, and 2.0 g/L PHA content with 0.5% rice bran after 24 h at 37 °C. PHA production yield obtained from *B. cereus* was 35.5%, 4.5 g/L CDM, and 1.6 g/L PHA content with 0.5% rice bran after 24 h of incubation at 37 °C. The maximum CDM and PHA content was obtained from *P. aeruginosa* after 72 h of incubation with 2 and 1.5% rice bran, respectively, 21.7 g/L and 3.8 g/L.

Under optimized conditions of sugarcane molasses, *P. aeruginosa* produced the maximum PHA yield of 95%, 1.9 g/L PHA content, and 2.0 g/L CDM in the presence of 2% sugarcane molasses after 96 h, at 37 °C. *Ali & Jamil (2017)* reported a 45.74% production yield of PHA, with 1.40 g/L CDM in the presence of glucose as the carbon source, which is less than the presented results in PHA production yield obtained from *Alcaligenes* sp. was 90.9%, 1.0 g/L PHA content and 1.1 g/L CDM with 8% cane molasses, after 72 h at 37 °C. PHA production yield obtained from *B. subtilis* was 91.6%, 1.1 g/L PHA content, and 1.2 g/L CDM with 2% cane molasses after 24 h at 37 °C. PHA production yield obtained from *B. cereus* was 80%, 1.2 g/L PHA content, and 1.5 g/L CDM with 2% cane molasses after 24 h at 37 °C. The maximum CDM and PHA content was obtained

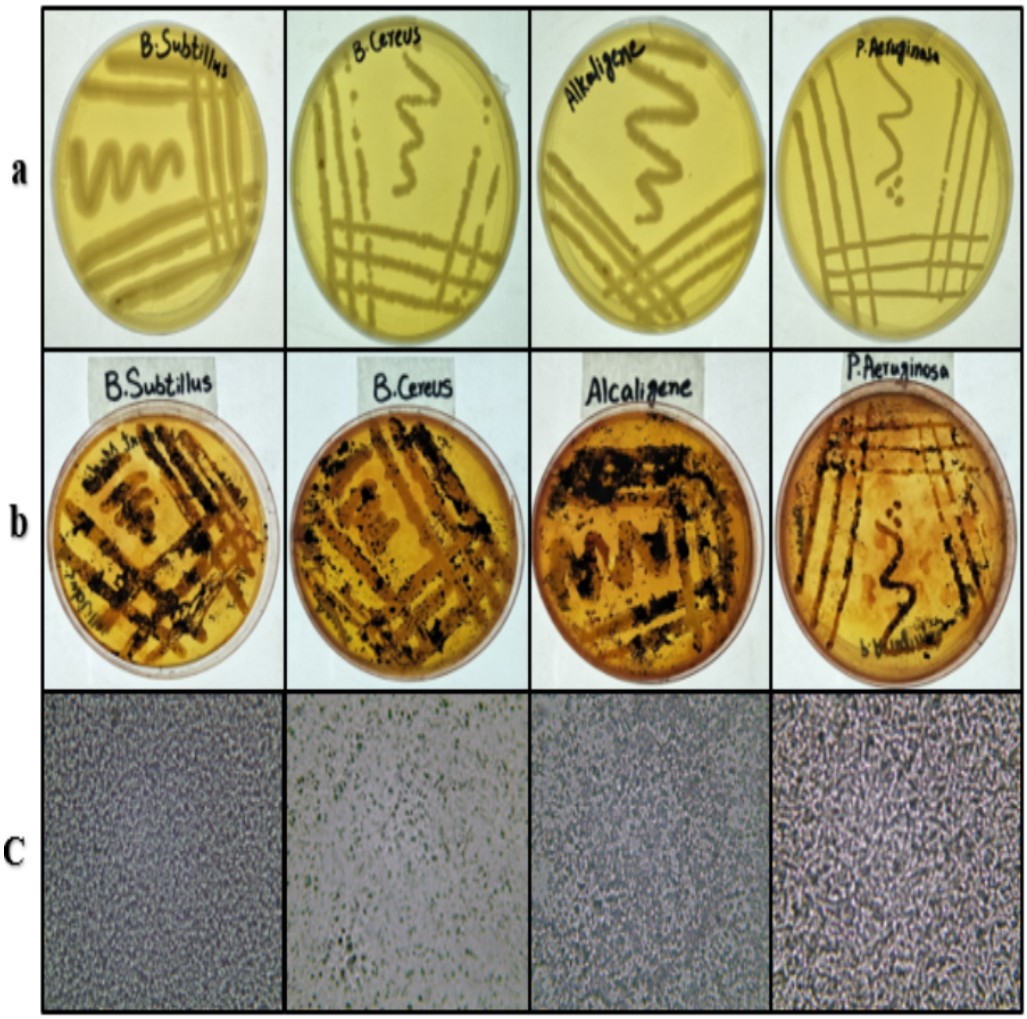

**Figure 1** **Bacterial strains cultured, stained and viewed under microscope.** From left to right, (A) streaking on nutrient agar, (B) Sudan Black-B staining and (C) microscopic images (left to right) of *B. subtilis*, *B. cereus*, *Alcaligenes sp.*, *P. aeruginosa*. Photographed by Ansa Naseem.

from *P. aeruginosa* after 96 h with 10% cane molasses, respectively, 5.9 g/L and 4.3 g/L. To find the optimum pH, the PHA production was studied from pH 4–9 (Figs. 3 and 4). PHA production was maximum at pH 8 with sugarcane molasses; in the case of rice bran, maximum PHA was produced at pH 9. Hence, *Alcaligenes sp.* effectively utilized the cane molasses for PHA production.

## FTIR spectroscopy

The functional groups of the extracted PHA from *B. subtilis*, *B. cereus*, *Alcaligenes* sp., and *P. aeruginosa* were analyzed by the FTIR spectrophotometer. Figure 5 indicated the presence of functional groups, which represented the more intense absorption band at 1,038.65 cm$^{-1}$, 1,056.51 cm$^{-1}$, 1,063.49 cm$^{-1}$ and 1,064.46 cm$^{-1}$ respectively, that described the -C-O- stretch of the ester group present in the structure of PHA. This

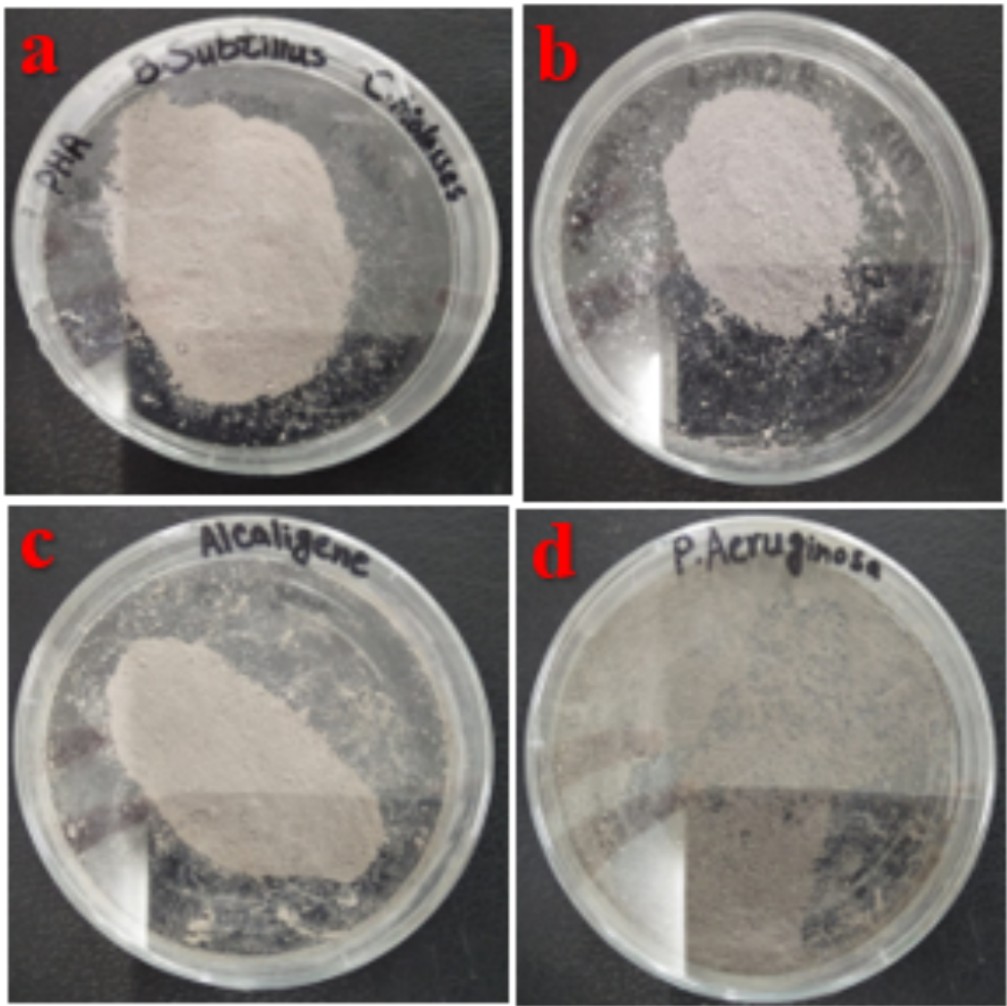

**Figure 2 Extracted PHA polymer.** PHA polymer extracted from the (A) *B. subtilis* (B) *B. cereus* (C) *Alcaligenes sp*. (D) *P. aeruginosa*. Photographed by Ansa Naseem.

absorption spectrum is similar to the previous studies in which the absorption band at 1,046 cm$^{-1}$, 1,053 cm$^{-1}$, and 1,043 cm$^{-1}$ indicates the presence of C-O carbonyl groups (*Mohapatra et al., 2017*; *Liu et al., 2021*; *Tyagi & Sharma, 2021*). Other absorbance peaks of specific characteristics are shown between the 1,500–2,000 (C =O stretch), 2,000–2,500 (-C ≡C- stretch), 2,500–3,000 (C-H stretch), 3,000-3,500 (O-H group), 600–1,000 (C-C stretch) which are similar to previous reports (*Maaloul et al., 2013*; *Abid, Raza & Hussain, 2016*). From the prior review of the literature and in the comparison to standard PHA, it can be concluded that the presence of these functional groups indicated the presence of poly-3-hydroxybutyrate (*Ojha & Das, 2018*; *Evangeline & Sridharan, 2019*). Hence the polymer extracted from *B. subtilis*, *B. cereus*, *Alcaligenes* sp., and *P. aeruginosa* was P(3HB), *i.e.,* poly-3-hydroxybutyrate.

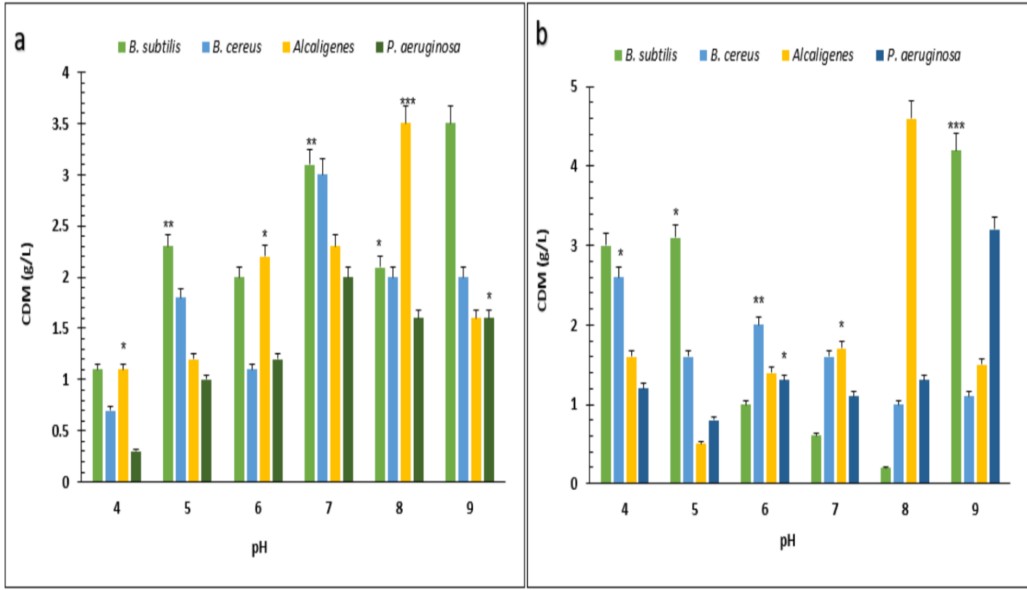

**Figure 3 Effect on the yield of PHA by using different carbon sources and pH values by various bacterial strains.** pH *versus* CDM (g/L) produced by *B. subtilis, B. cereus, Alcaligenes sp., P. aeruginosa* with (A) sugarcane molasses and (B) rice bran as sole carbon sources. The standard bars indicate the standard deviations of triplicate experiments. *$p < 0.05$, **$p < 0.03$, ***$p < 0.01$.

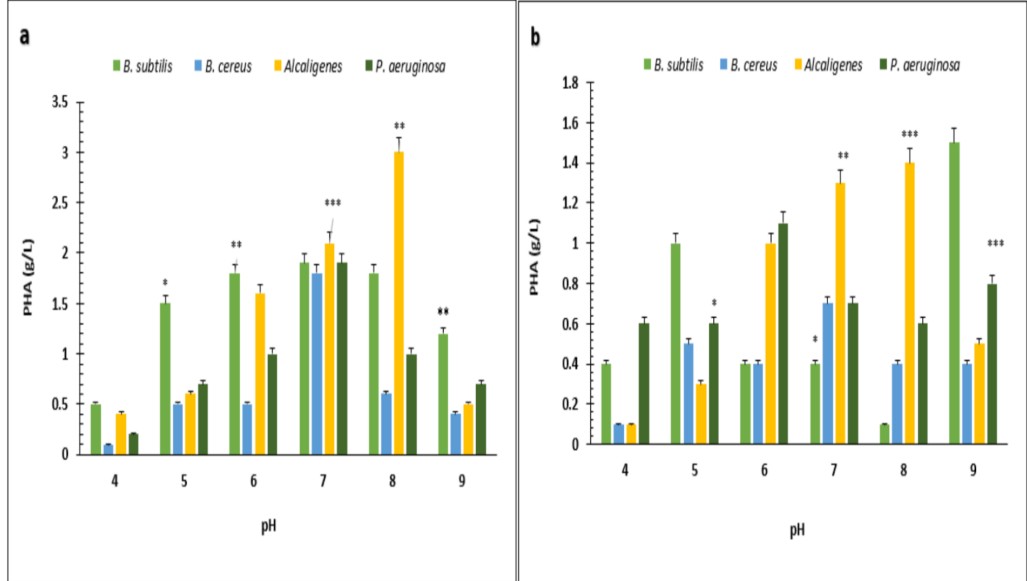

**Figure 4 Effect of pH on PHA yield produced by various bacterial strains.** pH *versus* PHA yield (g/L) produced by *B. subtilis, B. cereus, Alcaligenes sp., P. aeruginosa* with (A) sugarcane molasses and (B) rice bran as sole carbon sources. The standard bars indicate the standard deviations of triplicate experiments. *$p < 0.05$, **$p < 0.03$, ***$p < 0.01$.

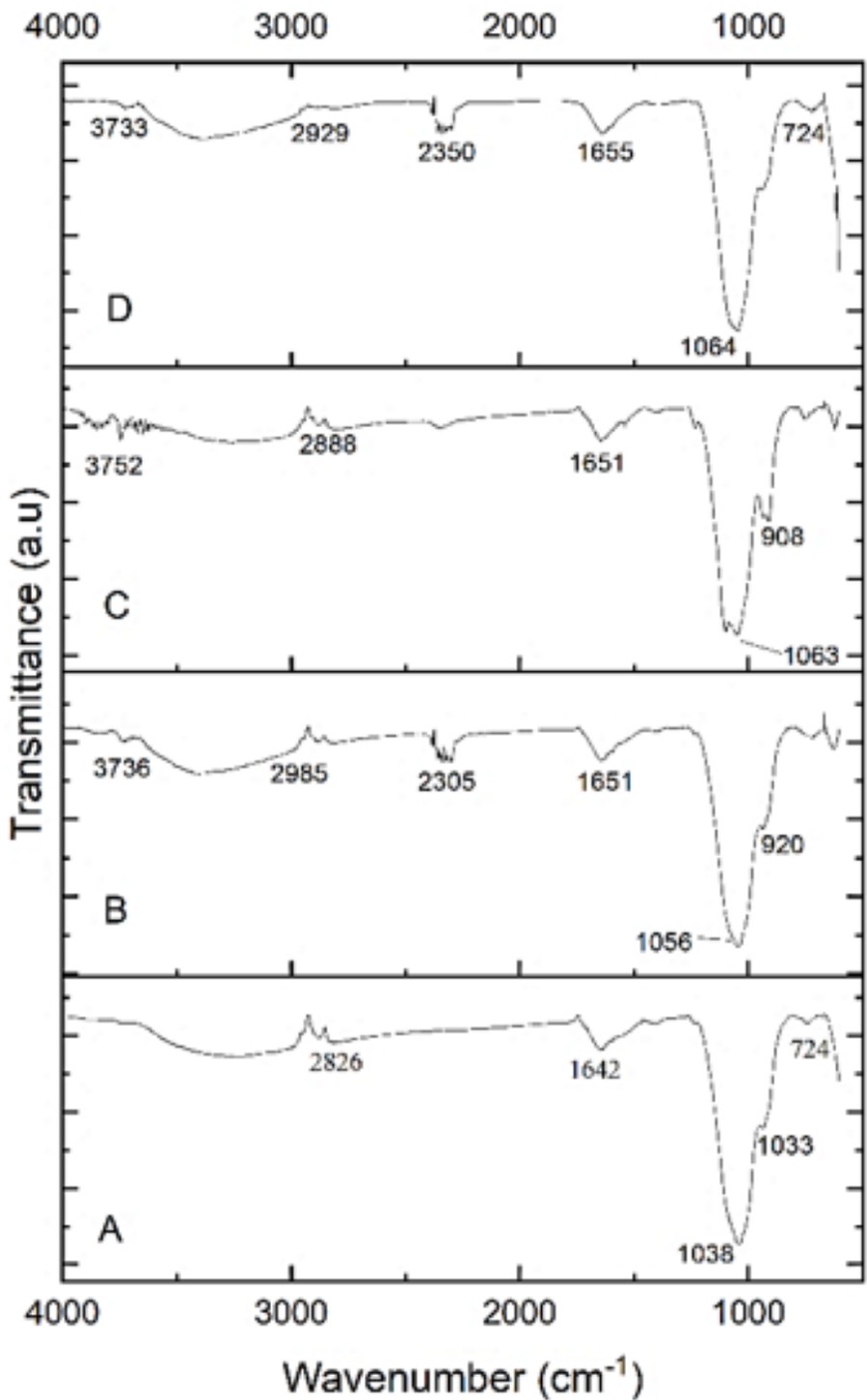

**Figure 5   FTIR characterization of PHA polymer.** FTIR analysis of the PHA polymer extracted from (A) *B. subtilis*, (B) *B. cereus*, (C) Alcaligenes sp., (D) *P. aeruginosa*.

## Differential scanning calorimetry (DSC) analysis

The PHA extracted from *B. subtilis, B. cereus*, *Alcaligenes* sp., and *P. aeruginosa* was thermally characterized by differential scanning calorimetry (DSC) to determine its melting temperature ($T_m$). DSC profiles of all four samples are shown in Fig. 6. PHA produced from *Bacillus subtilis* showed a sharp endothermic peak in the range of 160−170 °C, without significant change of overall heat absorption that represented its melting temperature ($T_m$). This data supported the previous reports (*Jung et al., 2020*; *Lorini et al., 2021*). In the case of *B. cereus* and *Alcaligenes* sp., no sharp peaks were observed, and the mild melting temperature curves were observed in the range of 130−150 °C and 230−240 °C. Recrystallization, crystal modification, and re-melting of thick crystals produce double endothermic peaks. This is similar to previous studies in which *B. megaterium* ATCC 14945 showed endothermic peaks at 133.15 and 145.42 °C (*Vu et al., 2021*; *Tian et al., 2021*). PHA produced from *P. aeruginosa* showed a sharp endothermic peak between the 230−240 °C range. Hence, the PHA produced during this study is thermally more stable at relatively high temperatures because melting temperature is the most important factor to describe the product quality and applications (*Xu et al., 2021*; *Karagöz, 2021*). The plastic for daily household applications, engineering, and high-temperature applications has a melting temperature of 100−300 °C, consistent with the PHA produced during this study and can be replaced with synthetic plastic (*Feldmann, 2016*).

## DISCUSSION

These experiments showed that B. *subtilis, B. cereus*, *Alcaligenes* sp., and *P. aeruginosa* strains produced PHA granules stained black due to Sudan Black-B under the compound microscope. Bacterial PHA was isolated from cellular biomass using sodium hypochlorite and chloroform digestion method. Under optimized conditions of rice bran, *P. aeruginosa* effectively utilized 0.5% rice bran and produced the maximum PHA yield, which was 93.7%, 3.2 g/L CDM, 3.0 g/L PHA content, after 72 h at 37 °C. According to previously reported studies, *Pseudomonas* sp. P (16) produced a 90.9% PHA yield (*Aljuraifani, Berekaa & Ghazwani, 2019b*), which has little difference from the research results presented in PHA production yield obtained from *Alcaligenes* sp. was 87.1%, 3.1 g/L CDM, and 2.7 g/L PHA content with 0.5% rice bran, after 72 h at 37 °C. PHA production yield obtained from *B. subtilis* was 69%, 2.9 g/L CDM, and 2.0 g/L PHA content with 0.5% rice bran after 24 h at 37 °C. PHA production yield obtained from *B. cereus* was 35.5%, 4.5 g/L CDM, and 1.6 g/L PHA content with 0.5% rice bran after 24 h of incubation at 37 °C. The maximum CDM and PHA content was obtained from *P. aeruginosa* after 72 h of incubation with 2 and 1.5% rice bran, respectively, 21.7 g/L and 3.8 g/L. Under optimized conditions of sugarcane molasses, *P. aeruginosa* produced the maximum PHA yield of 95%, 1.9 g/L PHA content, and 2.0 g/L CDM in the presence of 2% sugarcane molasses after 96 h, at 37 °C. *Ali & Jamil (2017)* reported a 45.74% production yield of PHA, with 1.40 g/L CDM in the presence of glucose as the carbon source, which is less than the presented results in PHA production yield obtained from *Alcaligenes* sp. was 90.9%, 1.0 g/L PHA content and 1.1 g/L CDM with 8% cane molasses, after 72 h at 37 °C. PHA production yield obtained from *B. subtilis* was

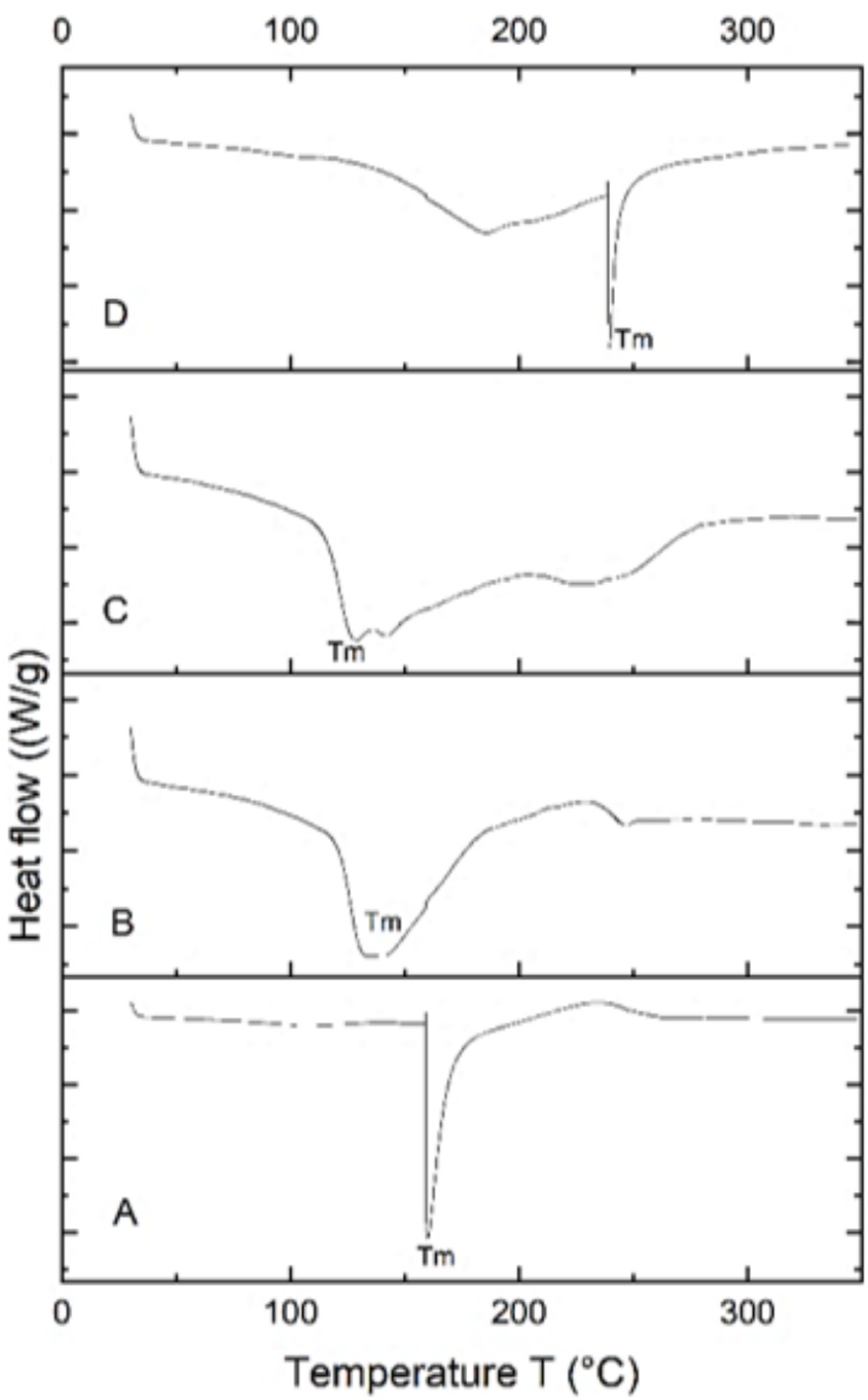

**Figure 6  Characterizing PHA polymer using DSC thermogram.** DSC thermogram of the PHA extracted from (A) *B. subtilis*, (B) *B. cereus*, (C) Alcaligenes sp. (D) *P. aeruginosa*.

91.6%, 1.1 g/L PHA content, and 1.2 g/L CDM with 2% cane molasses after 24 h at 37 °C. PHA production yield obtained from *B. cereus* was 80%, 1.2 g/L PHA content, and 1.5 g/L CDM with 2% cane molasses after 24 h at 37 °C. The maximum CDM and PHA content was obtained from *P. aeruginosa* after 96 h with 10% cane molasses, respectively, 5.9 g/L and 4.3 g/L. To find the optimum pH, the PHA production was studied from pH 4-9. PHA production was maximum at pH 8 with sugarcane molasses; in the case of rice bran, maximum PHA was produced at pH 9. Hence, *Alcaligenes sp.* effectively utilized the cane molasses for PHA production.

The functional groups of the extracted PHA from all the selected bacterial strains were analyzed by the FTIR spectrophotometer. The presence of functional groups, which represented the more intense absorption band at 1,038.65 cm$^{-1}$, 1,056.51 cm$^{-1}$, 1,063.49 cm$^{-1}$ and 1,064.46 cm$^{-1}$ respectively, described the -C-O- stretch of the ester group present in the structure of PHA. This absorption spectrum is similar to the previous studies in which the absorption band at 1,046 cm$^{-1}$, 1,053 cm$^{-1}$, and 1,043 cm$^{-1}$ indicates the presence of C-O carbonyl groups (*Mohapatra et al., 2017*; *Liu et al., 2021*; *Tyagi & Sharma, 2021*). Other absorbance peaks of specific characteristics are shown between the 1,500–2,000 (C =O stretch), 2,000–2,500 (-C ≡C- stretch), 2,500–3,000 (C-H stretch), 3,000–3,500 (O-H group), 600–1,000 (C-C stretch) which are similar to previous reports (*Maaloul et al., 2013*; *Abid, Raza & Hussain, 2016*). From the prior review of the literature and in the comparison to standard PHA, it can be concluded that the presence of these functional groups indicated the presence of poly-3-hydroxybutyrate (*Ojha & Das, 2018*; *Evangeline & Sridharan, 2019*). Hence the polymer extracted from *B. subtilis*, *B. cereus*, *Alcaligenes* sp., and *P. aeruginosa* was P(3HB), *i.e.,* poly-3-hydroxybutyrate. The PHA extracted from various these bacterial strains were thermally characterized by differential scanning calorimetry (DSC) to determine its melting temperature (T$_m$). PHA produced from *B. subtilis* showed a sharp endothermic peak in the range of 160−170 °C, without significant change of overall heat absorption that represented its melting temperature (T$_m$). This data supported the previous reports (*Jung et al., 2020*; *Lorini et al., 2021*). In the case of *B. cereus* and *Alcaligenes* sp., no sharp peaks were observed, and the mild melting temperature curves were observed in the range of 130−150 °C and 230−240 °C. Recrystallization, crystal modification, and re-melting of thick crystals produce double endothermic peaks. This is similar to previous studies in which *B. megaterium* ATCC 14945 showed endothermic peaks at 133.15 and 145.42 °C (*Vu et al., 2021*; *Tian et al., 2021*). PHA produced from *P. aeruginosa* showed a sharp endothermic peak between the 230−240 °C range. Hence, the PHA produced during this study is thermally more stable at relatively high temperatures because melting temperature is the most important factor to describe the product quality and applications (*Xu et al., 2021*; *Karagöz, 2021*). The plastic for daily household applications, engineering, and high-temperature applications has a melting temperature of 100−300 °C, consistent with the PHA produced during this study and can be replaced with synthetic plastic (*Feldmann, 2016*).

## CONCLUSION

This research was conducted for the cost-effective production of PHA from *B. subtilis*, *B. cereus*, *Alcaligenes* sp. and *P. aeruginosa* by using rice bran and sugarcane molasses as the carbon source. Increased concentrations of rice bran and sugarcane molasses decreased the PHA yield. All the selected bacterial strains utilized the cane molasses more effectively than rice bran. Hence, sugarcane molasses proved to be the best carbon source for the maximum production of PHA under optimized conditions of incubation period at pH 7 and 37 °C. The 24–48 h incubation period was best suited for full PHA production from *B. subtilis* and *B. cereus*, while the 72–96 h incubation period was best suited for maximum PHA production from *Alcaligenes* sp. and *P. aeruginosa*. FTIR and DSC analysis confirmed the production of P(3HB) and thermal stability of the produced P(3HB).

### Funding
The authors received no funding for this work.

### Competing Interests
The authors declare there are no competing interests.

### Author Contributions
- Aansa Naseem conceived and designed the experiments, performed the experiments, analyzed the data, prepared figures and/or tables, and approved the final draft.
- Ijaz Rasul performed the experiments, authored or reviewed drafts of the article, and approved the final draft.
- Zulfiqar Ali Raza performed the experiments, authored or reviewed drafts of the article, and approved the final draft.
- Faizan Muneer conceived and designed the experiments, analyzed the data, prepared figures and/or tables, and approved the final draft.
- Asad ur Rehman analyzed the data, authored or reviewed drafts of the article, and approved the final draft.
- Habibullah Nadeem conceived and designed the experiments, analyzed the data, authored or reviewed drafts of the article, and approved the final draft.

### Data Availability
Raw data is available as a Supplementary File.

### Supplemental Information
Supplemental information for this article can be found online at http://dx.doi.org/10.7717/peerj.17936#supplemental-information.

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
