# Peer review of "Bacterial production of polyhydroxyalkanoates (PHAs) using various waste carbon sources"

_PeerJ, doi:10.7717/peerj.17936_

## Round 0.1 · original submission · Major Revisions

Dear Dr. Naseem and colleagues:

Thanks for submitting your manuscript to PeerJ. I have now received two independent reviews of your work, and as you will see, the reviewers raised some concerns about the research. Despite this, these reviewers are optimistic about your work and the potential impact it will have on research studying bacterial polyhydroxyalkanoate production. Thus, I encourage you to revise your manuscript, accordingly, considering all the concerns raised by both reviewers.

Please provide more information to support your analyses, especially statistics. Please address this and ensure that all information regarding experimental design, gene and bacterial names, primers, etc. are provided and correct. Please strive to make your study repeatable. This especially includes providing all the necessary information in the Materials and Methods. All figures need to be clear and stand-alone.

There appears to be missing references that should be included within the framework of your study. Please enlist of a native English speaker to help with language and grammar. Also, take the advice of the reviewers for improving the clarity and organization of your manuscript.

There are other minor concerns raised by the reviewers that should all be addressed in your revision.

Please note that reviewer 2 has included a marked-up version of your manuscript.

I look forward to seeing your revision, and thanks again for submitting your work to PeerJ.

Good luck with your revision,

-joe

**Language Note:** PeerJ staff have identified that the English language needs to be improved. When you prepare your next revision, please either (i) have a colleague who is proficient in English and familiar with the subject matter review your manuscript, or (ii) contact a professional editing service to review your manuscript. PeerJ can provide language editing services - you can contact us at [email protected] for pricing (be sure to provide your manuscript number and title). – PeerJ Staff

·

Basic reporting

The manuscript titled "Bacterial Production of Polyhydroxyalkanoates (PHAs) Using Various Waste Carbon Sources (#95761)" delves into the economical production of PHA utilizing waste carbon sources, namely rice bran and sugarcane molasses, by Bacillus subtilis, Bacillus cereus, Alcaligenes sp., and Pseudomonas aeruginosa. Given the pressing environmental concerns surrounding synthetic plastics, the quest for biodegradable alternatives is paramount. Notably, the authors achieved a high yield of PHB%, highlighting the potential of their approach. The experimental design explanations provided were found to be insufficient in reaching the expressed conclusions. The identified shortcomings and uncertainties are outlined below.

Experimental design

Major Comments:
1. What is the glucose content of the sugarcane molasses and rice bran added to the fermentation media?
2. The rationale behind not obtaining hydrolysates of rice bran with enzymatic digestion should be elucidated. If this is attributed to the enzymes produced by PHA-producing microorganisms, it's crucial to express and conduct related enzymatic analyses to confirm their enzyme production potentials.
3. It's unclear from the text whether the authors conducted pH optimization after determining the optimum incubation time and % rice bran/sugarcane molasses concentration for maximum PHB production by the bacteria. Please clarify this matter.


Minor Comments:
1. Line 45: Insert a blank space between 0.02 and g/L.
2. Lines 102-104: Reframe the statement regarding PHA production of bacteria without using the term "living organisms."
3. Lines 157-159: Specify the five different concentrations of rice bran and pretreated sugarcane molasses added to the MSM medium within brackets.
4. Line 205: Use the abbreviated style of bacterial names throughout the text after their first mention with full names.
5. The resolution of Fig 1 is insufficient to compare staining of the strains with Sudan Black B.
6. The raw data file is not clear in the current format, please clarify it by using a Table option.
7. Lines 280-281: Italicize the bacterial names and maintain this style throughout the text.
8. Tables were not mentioned in the text. Please prove them.

Validity of the findings

4. It is important to incorporate statistic analysis to underscore the significance of the obtained data. It woluld be better to use a statistical program like GraphPad prism for rigorous evaluation of experimental data
5. How many replicates (biological or technical ) were used in the experiments? Please insert standart deviation values (±) in Table 1 and Table 2.

Additional comments

The manuscript requires substantial revisions. Once these revisions are completed, it can be considered for publication evaluation in PeerJ.

Reviewer 2 ·

Basic reporting

No comment

Experimental design

No comment

Validity of the findings

No comment

Additional comments

The referencing style is not according to PeerJ guidelines. Authors should consider a recently published article by PeerJ to have a better understanding of the referencing style to be followed.

Annotated reviews are not available for download in order to protect the identity of reviewers who chose to remain anonymous.

---

## Round 0.2 · Minor Revisions

Dear Dr. Naseem and colleagues:

Thanks for revising your manuscript. The reviewers are mostly satisfied with your revision (as am I). Great! However, there are a few remaining concerns to address. Please do not use bold formatting in the tables, and per reviewer 1, the related statistical method section needs to be included in the revised manuscript.

Please address these ASAP so we may move towards acceptance of your work.

Best,

-joe

·

Basic reporting

The revised manuscript has all the items that should be considered in the current form. The response of the authors is satisfactory. The statistical analysis is incorporated into the experiment. However, the related statistical method section should have been included in the revised manuscript.

Experimental design

no comment

Validity of the findings

no comment

Additional comments

no comment

Reviewer 2 ·

Basic reporting

evaluated already in revisions - Passed

Experimental design

evaluated already in revisions - Passed

Validity of the findings

evaluated already in revisions - Passed

---

## Round 0.3 · accepted · Accept

Dear Dr. Naseem and colleagues:

Thanks for revising your manuscript based on the concerns raised by the reviewers. I now believe that your manuscript is suitable for publication. Congratulations! I look forward to seeing this work in print, and I anticipate it being an important resource for groups studying bacterial polyhydroxyalkanoate production. Thanks again for choosing PeerJ to publish such important work.

Best,

-joe